Effect of modulating the extracellular matrix cross linkage by genipin on tumor cell resistance and survival in thioacetamide-induced hepatocellular carcinoma in rats

Hassan Hanan M. 1
Baradwan Mohammed 2
Bagalagel Alaa 3
Diri Reem 3
Basilim Ahmed 3
Nasrullah Mohammed Z. 4
Althagafi Abdulhamid 3
Kutbi Hussam I. 3
Mohammed Abdulaziz A. 3
Alshan Hanan M. 5
Al-Gayyar Mohammed M.H. mhgayyar@yahoo.com 6 7
1 Department of Pharmacology and Biochemistry, Delta University for Science and Technology , Gamasa , Egypt
2 Department of Medicinal Chemistry, King Abdul Aziz University , Jeddah , Saudi Arabia
3 Department of Pharmacy Practice, Faculty of Pharmacy, King Abdul Aziz University , Jeddah , Saudi Arabia
4 Department of Pharmacology and Toxicology, King Abdul Aziz University , Jeddah , Saudi Arabia
5 Faculty of Pharmacy, University of Tabuk , Tabuk , Saudi Arabia
6 Department of Biochemistry, Mansoura University , Mansoura , Egypt
7 Department of Pharmaceutical Chemistry, University of Tabuk , Tabuk , Saudi Arabia
Gould Gwyn
Electronic publication date: 2025 Jul 9
Publication date: 2025
Volume: 13
Electronic Location ID: e19680
Received 2025 Feb 10; Accepted 2025 Jun 10
Copyright: ©2025 Hassan et al.
Copyright year: 2025
Copyright holder: Hassan et al.
License: This is an open access article distributed under the terms of the Creative Commons Attribution License, which permits unrestricted use, distribution, reproduction and adaptation in any medium and for any purpose provided that it is properly attributed. For attribution, the original author(s), title, publication source (PeerJ) and either DOI or URL of the article must be cited.
License URL: https://creativecommons.org/licenses/by/4.0/

Keywords: Epidermal growth factor (EGF), Epidermal growth factor receptor (EGFR), Fibronectin, Genipin, Glycogen synthase kinase-3 beta (GSK-3β), Hepatocellular carcinoma, Protein kinase B (PKB), Versican

Funding: The authors received no funding for this work.

==============================
Background

Previous studies on patients and rats with hepatocellular carcinoma (HCC) have shown significant changes in the extracellular matrix (ECM). Versican, a component of the ECM, forms extensive multimolecular interactions with other ECM components, particularly hyaluronan, through specific domains in its core protein. However, disturbances in the hyaluronan-versican interaction may affect cancer development. We aimed to examine the effect of modulating matrix cross-linkage between hyaluronan and versican using genipin on tumor cell survival, resistance, and renewal.

Methods

Following the induction of HCC in rats using thioacetamide, an oral dose of 10 mg/kg of genipin was administered. Liver impairment was evaluated by measuring serum α-fetoprotein (AFP) levels and examining liver sections stained with hematoxylin/eosin and anti-versican and anti-fibronectin antibodies. Additionally, hepatic expression levels of mRNA and proteins, including epidermal growth factor (EGF), epidermal growth factor receptor (EGFR), fibronectin, glycogen synthase kinase-3 beta (GSK-3β), protein kinase B (PKB), and versican, were analyzed.

Results

Genipin enhances rats’ survival, leading to reduction in serum AFP levels and number of hepatic nodules. Micro-imaging examinations reveal that genipin reduces vacuolated cytoplasm, apoptotic nuclei, and necrotic nodules. Additionally, it significantly lowers EGF, EGFR, fibronectin, GSK-3β, PKB, and versican expression levels.

Conclusion

Genipin may be considered novel anticancer agent with hepatoprotective effects. This is achieved by reducing versican-free forms. Additionally, genipin decreases tumor cells’ resistance by lowering the expression of EGF, EGFR, PKB, and GSK-3β. Finally, it reduces tumor cell survival by decreasing the expression of fibronectin.

Introduction

Hepatocellular carcinoma (HCC) constitutes the fifth most prevalent cancer globally and stands as the third leading cause of cancer-related mortality (Ferlay et al., 2019). HCC is responsible for approximately 90% of all primary liver cancer, with nearly one million new HCC diagnoses reported annually, resulting in approximately 600,000 fatalities annually (Gomaa et al., 2008). The prognosis for patients diagnosed with liver cancer remains poor, primarily due to late-stage detection, resistance to existing therapies, and a high recurrence rate. The overall five-year survival rate for patients with liver cancer is merely 18%, while the recurrence rate exceeds 50% in five years following surgical intervention (Sas et al., 2022). Cirrhosis is recognized as a significant independent risk factor for the development of HCC worldwide, with nearly 90% of individuals diagnosed with HCC presenting with underlying cirrhotic liver disease (West et al., 2017). Although there is increasing interest in investigating therapeutic targets for HCC, effective treatment options remain significantly limited (Manzar et al., 2022). Furthermore, the tumor’s considerable capacity to invade blood vessels further constrains the available treatment alternatives for individuals suffering from HCC (Ferlay et al., 2019).

In patients diagnosed with early-stage HCC and possessing adequate liver function, surgical resection and local ablation represent common curative interventions. Unfortunately, a significant number of HCC patients present at advanced stages, precluding the possibility of surgical resection or liver transplantation (Wege, Li & Ittrich, 2019). Currently, the only FDA-approved targeted therapies for HCC are tyrosine kinase inhibitors, sorafenib as the first-line treatment option and Regorafenib as the second-line alternative. Sorafenib has been associated with a modest enhancement in overall survival, offering patients no more than an additional three months of life. Common adverse effects, including diarrhea, fatigue, and hand-foot skin syndrome, are prevalent in approximately 80% of cases, leading to discontinuation of the drug in about 20% of patients (Marrero et al., 2016). Similarly, regorafenib provides limited benefit, extending the survival of patients who have developed resistance to Sorafenib by an additional two to three months. Recently, nivolumab, an immune checkpoint inhibitor that targets the inhibitory T-cell receptor programmed death 1 (PD-1), received FDA approval for use as a second-line treatment; however, it exhibits an overall tumor response rate of only 20% among patients with HCC (El-Khoueiry et al., 2017).

The liver microenvironment comprises various components, including the extracellular matrix (ECM), immune cells, Kupffer cells, endothelial cells, fibroblasts, cytokines, and various growth factors (Hernandez-Gea et al., 2013). The ECM is a complex three-dimensional network composed of diverse macromolecules. It encases stromal, endothelial, and epithelial cells, providing essential structural support for these cellular and tissue components. Primarily constituted of glycosaminoglycans and collagen, the ECM establishes the requisite framework for cell attachment, while glycosaminoglycans occupy this framework (Abdel-Hamid, 2009). The altered expression of proteoglycans on tumor and stromal cells significantly impacts cancer cell signaling, resulting in enhanced tumor growth, migration, and angiogenesis (Theocharis et al., 2010). Versican, a prominent member of the chondroitin sulfate proteoglycan family, is characterized by a C-terminal G3 domain that encompasses two epidermal growth factor (EGF)-like repeats, a complement binding protein (CBP)-like motif, and a C-type lectin-like motif (LeBaron, Zimmermann & Ruoslahti, 1992). Consequently, versican contributes to various processes occurring at the tumor site, including cellular adhesion, migration, proliferation, apoptosis, and angiogenesis.

Genipin is a natural iridoid aglycone extracted from the Chinese herb Gardenia jasminoides, which has been extensively utilized in the management of conditions associated with inflammation and heat-induced liver cancer (Wu et al., 2023). Numerous studies have emphasized the therapeutic properties of genipin in various inflammatory disorders, including acute liver injury, severe hepatitis, and acute lung injury, while also providing protective benefits to both cardiovascular and nervous systems (Liu et al., 2024). Furthermore, genipin is recognized as a widely employed biocompatible cross-linking agent, attributed to its excellent biodegradability, remarkable biocompatibility, and stable cross-linking characteristics (Yu et al., 2021). In addition, genipin has been shown to effectively cross-link the extracellular matrix (ECM), significantly enhancing its mechanical stiffness and structural integrity. This modification leads to a reorganization of the actin cytoskeleton, facilitating improved cellular adhesion and communication. Furthermore, genipin promotes the transcription of genes that are dependent on yes-associated protein, which are crucial for cellular growth, proliferation, and survival, thereby influencing various physiological and pathological processes (Saito-Diaz et al., 2024). In this context, our research aims to explore a novel pathway for developing HCC through the interaction between hyaluronan and versican within ECM. We intend to examine the impact of genipin on tumor cell survival, resistance, and renewal.

Materials & Methods

Experimental design

Forty Sprague-Dawley rats, encompassing male and female specimens with weights ranging from 180 to 200 grams, were used. They were procured from the college’s animal facility and had not undergone any prior procedures. They were maintained in a controlled environment with a stable temperature and were subjected to a 12-hour light and dark cycle. The rats were allowed a five-day acclimatization period. The Research Ethics Committee of the Faculty of Pharmacy at Delta University of Science and Technology approved the animal research protocol (FPDU8/2023). The rats were afforded unrestricted access to standard rat chow and water throughout the experiment. They were categorized into four groups containing ten rats, with two housed per cage. All treatments administered to the rats were performed by a single research team member, following a consistent sequence and schedule during the morning hours. To minimize animal pain during experiments, five mg/kg of oral diclofenac sodium is administered to the HCC rats to alleviate discomfort effectively. In instances where the animals experience severe pain or show signs of significant distress, such as a weight loss exceeding 30% of their baseline weight, humane euthanasia is performed to prevent further suffering. The rats were randomly assigned to four groups, keeping ten rats in each group, using http://www.randomizer.org:

Control group: Rats were administered normal saline daily for sixteen weeks.

Treated control group: Rats received 10 mg/kg of genipin (Sigma Aldrich Chemicals Co, St. Louis, MO) orally using oral gavage in normal saline once daily for sixteen weeks.

HCC group: Rats were given 200 mg/kg of thioacetamide in normal saline via intraperitoneal injection twice weekly for sixteen weeks (Abdulal et al., 2024; Albalawi et al., 2024; Alshehri et al., 2024; El-Far et al., 2020).

HCC and genipin group: Rats were administered 200 mg/kg of thioacetamide intraperitoneally in normal saline twice a week, along with 10 mg/kg of genipin orally using oral gavage in normal saline daily for sixteen weeks.

Sample collection

Thiopental sodium (40 mg/kg, intraperitoneal injection) was utilized to induce anesthesia in the rats. Serum samples were collected from the retro-orbital plexus and subsequently centrifuged at 3,000 rpm for five minutes. Rats were euthanized by cervical dislocation then the entire liver was separated. A section of the liver tissue containing tumors was carefully excised, sliced, and preserved in a 10% (w/v) buffered formalin solution for morphological examination. Furthermore, an additional portion of the hepatic tumor tissue was homogenized in sodium-potassium phosphate buffer (pH 7.4) at a ratio of 1:10 and subsequently stored at −80 °C as described previously by our group (Hassan et al., 2024).

Liver gross appearance and evaluation of the number of hepatic nodules

The macroscopic characteristics of the harvested liver, including color, size, texture, and the presence of nodules, were documented. Abnormal nodules were distinguished by their subtle grayish-white coloration. The entire liver tissue was excised, preserved, and subsequently sliced into 2-mm-thick sections. Two independent investigators counted nodules within the liver that measured three mm or greater, as described previously by our group (Hassan et al., 2024).

Histopathological examination and immunohistochemistry

Hepatic slices preserved in 10% formalin were processed according to standard histopathological techniques and subsequently embedded in paraffin blocks, sectioned to a thickness of five micrometers. The sections were stained with hematoxylin and eosin. Each section was assigned an anonymous code and evaluated blindly using a digital camera-based imaging system (Nikon Corporation). For immunohistochemistry, the paraffin sections, also five micrometers thick, were subjected to treatment with monoclonal antibodies targeting versican and fibronectin (MyBioSource, San Diego, CA, USA). A counterstain of hematoxylin was applied to the sections. All readings were conducted by a pathologist who remained unaware of the treatment groups (Albalawi et al., 2023; Bayomi et al., 2013; Elsherbiny, Al-Gayyar & Abd El Galil, 2015; Hassan et al., 2024).

Enzyme-linked immunosorbent assays (ELISA) determination

The concentrations of α-fetoprotein (AFP), epidermal growth factor (EGF), epidermal growth factor receptor (EGFR), fibronectin, glycogen synthase kinase-3 beta (GSK-3β), protein kinase B (PKB), and versican were analyzed utilizing commercially available enzyme-linked immunosorbent assay (ELISA) kits. These kits were sourced from (MyBioSource, San Diego, CA, USA), ensuring reliable results for the study.

Quantitative real-time polymerase chain reaction (RT-PCR)

In the central lab of the faculty, the procedure was conducted as described previously by our group (Alattar, Alshaman & Al-Gayyar, 2022; Alharbi et al., 2024; Bagalagel et al., 2022b; Hassan et al., 2024). Total RNA was extracted utilizing the RNeasy Mini kit (Qiagen, Valencia, CA, USA), with the concentration assessed using Maxima® SYBR Green/Fluorescein Master Mix (Fermentas). Subsequently, 1 µg of RNA was reverse transcribed into complementary DNA (cDNA) employing the QuantiTect® Reverse Transcription Kit (Qiagen). The expression levels of mRNA for EGF, EGFR, fibronectin, GSK-3β, PKB, and versican in rat hepatic lysates were quantified using Maxima® SYBR Green/Fluorescein qPCR Master Mix in conjunction with the Rotor-Gene Q (Qiagen). The primer set used in the study was summarized in Table 1. The thermal profile of RT-qPCR is reverse transcriptase 1 cycle at 55 °C for 10 min, followed by enzyme inactivation at 59 °C for 2 min, then forty cycles of amplification using 95 °C for 10 s, 55 °C for 10 s, and finally 72 °C for 30 s. Final extension is done at 72 °C for 5 min. For housekeeping and internal referencing, rat glyceraldehyde 3-phosphate dehydrogenase (GAPDH) was utilized. The outcomes of the RT-PCR analysis were represented as cycle threshold (Ct) values. The PCR data sheet provides the Ct values for the genes of interest about the reference housekeeping gene, GAPDH. A control sample was included to evaluate the gene expression of each specific gene. The relative quantification (RQ) for each target gene was computed and standardized to the housekeeping gene using the delta-delta Ct (ΔΔCt) method. The RQ of each gene was determined with formula 2-ΔΔCt.

Table 1 Primer set used.

Primer	Sequence	Location	Length	Accession number	
Versican	Forward	5′- GCTTTAAACGACCTGATCTCTGC-3′	1,314	318	NM_001170559.1	
Reverse	5′-CATCCCATGTACGGCGATGA-3′	1,631	
EGF	Forward	5′-ACGACCTACAATTGGACCGA-3′	1,778	936	NM_012842.2	
Reverse	5′-TTTTCCCATCTGGGGCCTTC-3′	2,713	
EGFR	Forward	5′-ACCGTGTGGGAACTGATGAC-3′	2,949	313	NM_031507.2	
Reverse	5′-TGTCCTCCTCCTCCATCAGG-3′	3,261	
PKB	Forward	5′-TCATTGAGCGCACCTTCCAT-3′	286	364	NM_033230.3	
Reverse	5′-TTCTGCAGGACACGGTTCTC-3′	649	
GSK-3β	Forward	5′-AGCTGATCTTTGGAGCCACC-3′	816	661	NM_032080.1	
Reverse	5′-AACGTGACCAGTGTTGCTGA-3′	1,476	
fibronectin	Forward	5′-ACAGGTCATCTTTGGCTTCTGT-3′	1,141	241	AH002173.2	
Reverse	5′-TTGGCCGTTTCAGGAAGGTT-3′	1,381	
GAPDH	Forward	5′-GCGAGATCCCGCTAACATCA-3′	306	944	NM_017008.4	
Reverse	5′ATTCGAGAGAAGGGAGGGCT-3′	1,249	

Statistical analysis

The data are presented as mean ± SEM. The Kolmogorov–Smirnov test was utilized to evaluate the normality of the sample distribution. The number of samples in each experiment of gene expression and hepatic protein levels is equal to ten rats. The Kaplan–Meier method was employed to analyze significant differences in survival rates. A one-way ANOVA, followed by the Bonferroni post hoc test, was performed to identify significant differences between groups. Statistical analyses were conducted using SPSS version 22 (IBM Corp.), with a significance threshold set at p < 0.05.

Results

Antitumor effects of genipin on rats

The survival rate of the control groups was consistently recorded at 16 weeks, with no significant changes noted in the treated control groups. Conversely, the group with HCC exhibited a markedly reduced survival rate of only 20%. However, upon administration of genipin, the survival rate for the HCC group significantly improved, rising to an impressive 90%. Further analysis indicated that the average relative serum AFP level in HCC group was approximately 6.86 times higher than that of the control group. Following treatment with genipin, the elevated serum AFP levels substantially decreased, with a reduction of approximately 73.31%. Moreover, the HCC group displayed a significant increase in the number of hepatic nodules compared to the control group, providing additional evidence of disease progression. In contrast, the group treated with genipin exhibited a reduction of roughly 71.43% relative to the untreated HCC group. These findings are visually represented in Fig. 1.

Figure 1 Effect of HCC and 10 mg/kg genipin on rats’ survival (A), representive liver images (B), the average number of nodules (C), and serum AFP (D).

* Significant difference as compared with the control group at p < 0.05. # Significant difference as compared with the HCC group at p < 0.05. AFP, alpha-fetoprotein; HCC, hepatocellular carcinoma.

Effect of genipin on the histology of the liver tissue

Microscopic examination of liver sections from the control groups, stained with H&E, reveals a typical architectural arrangement characterized by well-organized hepatic cords surrounding the central veins. Conversely, the liver sections obtained from the group with HCC exhibit features indicative of well-differentiated HCC. The neoplastic cells in this cohort display a polygonal morphology and possess distinct cell membranes with a cytoplasm that is notably eosinophilic and granular, suggesting a high protein content. Furthermore, numerous cells exhibit vacuolation, and apoptotic nuclei marked with black arrows indicate apoptotic cells. Additionally, necrotic nodules, highlighted by a yellow arrow, are apparent, reflecting the pathological changes associated with HCC. In liver sections from the HCC group that received treatment with genipin, a significant improvement in the structural integrity of hepatocytes is observed. This finding suggests a potential therapeutic effect of genipin on the histological architecture of the HCC liver (Fig. 2).

Figure 2 Hepatic sections stained with hematoxylin and eosin in the control group (A), control group treated with genipin (B), hepatocellular carcinoma (A), and hepatocellular carcinoma treated with genipin (D).

Images show vacuolated cytoplasm with apoptotic nuclei (black arrows) and necrotic nodules (yellow arrows). Scale bars represent 100 µm.

Effect of genipin and HCC on the expression of versican

The study identified a significant upregulation of versican gene expression in the liver, with a remarkable increase of 4.38-fold. In parallel, the levels of versican protein demonstrated a substantial increase, measured at 4.02-fold. Histological analysis of liver sections, utilizing anti-versican antibodies for staining, revealed a notable expansion in the stained areas. Conversely, treatment with genipin reversed these effects. Importantly, this treatment did not influence the control groups, suggesting a targeted effect of genipin on the altered gene and protein expression associated with HCC (Fig. 3).

Figure 3 Effect of HCC and 10 mg/kg genipin on hepatic gene expression of versican (A) and its hepatic protein level (B). Hepatic sections stained with anti-verscian antibodies in the control group (C), control group treated with genipin (D), HCC group (E), and HC.

* Significant difference as compared with the control group at p < 0.05. # Significant difference as compared with the HCC group at p < 0.05. The number of samples in each experiment of gene expression and hepatic protein levels is equal to ten rats. Scale bar 100 µm. HCC, hepatocellular carcinoma.

Effect of genipin and HCC on the expression of EGF and EGFR

We identified a significant upregulation in the expression levels of EGF and its receptor EGFR. Specifically, the expression of EGF was approximately 3.64 times greater, while EGFR expression increased by approximately 3.42 times compared to the control group. Furthermore, EGF and EGFR levels within the hepatic tissues exhibited notable elevations, with EGF levels rising by 3.08-fold and EGFR levels by 3.84-fold relative to the control group. Notably, the administration of genipin demonstrated to reverse the elevated expression levels of both EGF and EGFR in the HCC-affected rats, with no substantial alterations observed in the control rats receiving genipin treatment (Fig. 4).

Figure 4 Effect of HCC and 10 mg/kg genipin on hepatic gene expression of EGF (A) and EGFR (C) as well as protein expression of EGF (B) and EGFR (D).

* Significant difference as compared with the control group at p < 0.05. # Significant difference as compared with the HCC group at p < 0.05. The number of samples in each experiment of gene expression and hepatic protein levels is equal to ten rats. HCC, hepatocellular carcinoma; EGF, epidermal growth factor; EGFR, epidermal growth factor receptor.

Effect of genipin and HCC on the expression of PKB and GSK-3β

In investigating liver tissues from HCC, a significant increase in the expression levels of PKB and GSK-3β, was observed. Specifically, the expression of the PKB gene exhibited an increase by a factor of 2.69, whereas GSK-3β demonstrated an impressive rise of 3.87 times when compared to the control group. Furthermore, the analysis of hepatic tissue samples indicated that the protein levels of PKB and GSK-3β were also markedly elevated, with increases of 3.09 times and 4.21 times, respectively. Of particular note, administering proved effective in mitigating these alterations in the HCC-affected rats and did not impact the control group (Fig. 5).

Figure 5 Effect of HCC and 10 mg/kg genipin on hepatic gene expression of PKB (A) and GSK-3β (C) as well as protein expression of PKB (B) and GSK-3β (C).

* Significant difference as compared with the control group at p < 0.05. # Significant difference as compared with the HCC group at p < 0.05. The number of samples in each experiment of gene expression and hepatic protein levels is equal to ten rats. HCC, hepatocellular carcinoma; GSK-3β, glycogen synthase kinase-3 beta; PKB, protein kinase B.

Effect of genipin and HCC on the expression of fibronectin

The research findings indicated a significant increase in the expression of the fibronectin gene in the liver of HCC rats, quantified at an impressive 4.84-fold and a 3.36-fold increase in its protein level. Histological analysis of liver sections, utilizing anti-fibronectin antibodies for immunostaining, revealed a marked enlargement in the stained areas in HCC rats. Conversely, treatment with genipin effectively reversed these changes, without any effect on the control groups (Fig. 6).

Figure 6 Effect of HCC and 10 mg/kg genipin on hepatic gene expression of fibronectin (A) and its hepatic protein level (B). Hepatic sections stained with anti-fibronectin antibodies in the control group (C), control group treated with genipin (D), HCC group (E).

* Significant difference as compared with the control group at p < 0.05. # Significant difference as compared with the HCC group at p < 0.05. The number of samples in each experiment of gene expression and hepatic protein levels is equal to ten rats. Scale bar 100 µm. HCC, hepatocellular carcinoma.

Discussion

HCC represents a prevalent form of liver cancer, currently identified as the sixth most common cancer worldwide and the fourth leading cause of cancer-related mortality. The prognosis for patients diagnosed with HCC is concerning, characterized by an estimated five-year survival rate of approximately 18%. Chemotherapy is one of the primary treatment modalities for HCC; however, while it can effectively target neoplastic cells, it frequently induces significant physical and emotional changes, impairing patients’ overall quality of life. Furthermore, chemotherapy is associated with a variety of severe side effects and entails considerable financial implications (Bagalagel et al., 2022a). We conducted an experiment wherein HCC was induced in rats by administering thioacetamide, a chemical known for its carcinogenic properties associated with liver cancer. Thioacetamide resulted in a drastic decrease in the survival rate, with only 20% of the treated rats surviving throughout the observation period. Additionally, we observed a troubling increase in the average number of liver nodules and elevated serum levels of AFP. Histological examination of liver tissue microsections, stained with hematoxylin/eosin, revealed vacuolated cytoplasm with apoptotic nuclei and necrotic nodules. In conjunction with these findings, we also examined the effects of genipin, a naturally occurring compound recognized for its diverse pharmacological properties. The results demonstrated that treatment of HCC-induced rats with genipin led to a significant increase in survival rates and a noteworthy reduction in liver nodules and serum AFP levels following treatment. Furthermore, histological analysis of liver tissue microsections from genipin-treated HCC rats, similarly stained with hematoxylin/eosin, showed substantial improvements in hepatic tissue architecture. These findings suggest that genipin may represent a promising therapeutic agent in treating HCC, warranting further investigation into its mechanisms and potential clinical applications.

Genipin is widely recognized for its significant role in modulating the cross-linking of ECM. This compound is extensively employed as a biocompatible cross-linking agent, attributed to several critical properties. One of its foremost advantages is its exceptional biodegradability, which ensures safe decomposition within biological systems without inflicting harmful side effects (Nyambat et al., 2020). Furthermore, genipin demonstrates remarkable biocompatibility, rendering it suitable for various medical applications, particularly in tissue engineering and regenerative medicine. Its stable cross-linking characteristics enhance its efficacy in providing structural integrity and resilience to biomaterials, thereby improving performance across various applications (Utami Nike et al., 2022).

Versican, an essential extracellular matrix component, is integral to various biological processes, including cell adhesion, proliferation, migration, and angiogenesis. Its expression is significantly elevated in several malignancies, including liver cancer, lung cancer, breast cancer, gastric cancer, colon cancer, ovarian cancer, and bladder cancer. This increase in versican levels plays a critical role in the mechanisms that underpin malignant transformation and tumor progression (Xiong et al., 2023). Versican impacts fundamental aspects of tumor behavior, notably the proliferation of tumor cells, their invasion into adjacent tissues, and metastatic spread to distant locations. Furthermore, versican is involved in angiogenesis, the new blood vessel formation process that provides tumors with essential nutrients and oxygen, and regulates apoptosis which is crucial for tumor survival (El-Far et al., 2022). In this research study, we noted a significant increase in versican expression in rat models diagnosed with HCC. Notably, this elevated expression was effectively reduced following treatment with genipin. This finding is crucial as it represents the first documented study instance illustrating genipin’s ability to increase the cross-linkage between versican and hyaluronan, leading to a decrease in the free versican in the hepatic tissues.

Much new research has illustrated the role of EGF in the metastasis of primary tumors. EFG is believed to enhance both the tumorigenic potential and the antiangiogenic properties of HCC cells. Specifically, EGF may activate various signaling pathways that contribute to cellular growth, invasion, and the overall progression of cancer. Furthermore, it enhances angiogenesis in the tumor microenvironment, promoting tumor survival and expansion. These findings emphasize the significance of EGF in elucidating the complex mechanisms underlying HCC metastasis and highlight its potential as a viable therapeutic target in cancer treatment (Yan et al., 2020). EGFR is a prominent member of the ERBB family, comprising a group of transmembrane receptor tyrosine kinases (RTKs) essential for various cellular processes. Upon binding to its specific ligands, EGFR initiates a sequence of transphosphorylation, wherein the EGFR dimer activates itself by phosphorylating tyrosine residues. This mechanism facilitates the activation of several downstream intracellular signaling pathways (Zhang et al., 2024). However, dysregulation of EGFR signaling can result in uncontrolled cellular growth and cancer progression. It may arise from various factors, including gene amplification, activating mutations within the EGFR gene, or the receptor’s capacity to evade degradation. These alterations are commonly observed in multiple cancer types, including lungs, breast, glioblastoma, and liver cancers (Lim et al., 2023). Given the critical role of EGFR in cancer biology, targeting this receptor has emerged as a promising therapeutic strategy, particularly for patients diagnosed with HCC, in which EGFR overexpression or hyperactivation is frequently observed. Our research study identified a significant upregulation of EGF and its receptor EGFR in HCC rats. Notably, our findings indicate that the administration of genipin resulted in a substantial reduction in the levels of both EGF and EGFR. This represents a significant advancement, as it constitutes the first documented evidence in the literature illustrating genipin’s ability to inhibit the expression of EGF and its receptor in an animal model of HCC.

Akt, also known as protein kinase B (PKB), is a pivotal regulator of numerous cellular and biological processes. It plays a critical role in inhibiting apoptosis, essential for maintaining tissue homeostasis. Moreover, PKB is essential in stabilizing the genome, aiding in the prevention of mutations, and preserving the integrity of genetic material (Somade et al., 2024). In addition, Akt facilitates protein synthesis by promoting mRNA translation. It is also involved in glucose metabolism by enhancing insulin signaling pathways, ensuring cells have sufficient energy to support their survival and proliferation (Zhong et al., 2023). The activation of PKB is typically initiated by binding growth factors and other extracellular signals to their corresponding receptors located on the cell surface. This interaction triggers a cascade of intracellular signaling events that ultimately lead to the activation of PKB. However, a significant molecular characteristic observed in various human malignancies is the abnormal hyperactivation of the Akt/PKB pathway (He et al., 2021). This excessive activation frequently contributes to tumor aggressiveness. It enhances the resistance of cancer cells to chemotherapy and other targeted therapies. In this study, we observed a significant increase in the expression of PKB in rats, which increased with HCC. Notably, this elevated expression was substantially reduced following treatment with genipin, which has previously been documented to decrease levels of Akt/PKB in various cancer types, including osteosarcoma (Huang et al., 2023), oral squamous cell carcinoma (Wei et al., 2020), and bladder cancer (Li et al., 2018). However, it represents the first instance in which genipin has been demonstrated to inhibit Akt/PKB signaling in HCC.

GSK-3β is a well-characterized enzyme and a prominent downstream target within the Akt signaling pathway. It is essential in various cellular processes, including metabolism, cell proliferation, and survival. GSK-3β possesses two critical phosphorylation sites: serine 9 (Ser9) and tyrosine 216 (Tyr216). Phosphorylation at the Ser9 site inactivates GSK-3β, consequently diminishing its kinase activity. In contrast, phosphorylation at the Tyr216 site enhances the enzyme’s activity, amplifying its role in cellular signaling and metabolic regulation (Shao et al., 2023). The findings of GSK-3β in tumor biology often present a complex and sometimes contradictory. Numerous studies propose GSK-3β as a potential tumor suppressor. For instance, it has been reported that GSK-3β expression is markedly reduced in breast cancer tissues relative to normal tissues. Furthermore, the overexpression of GSK-3β has been demonstrated to augment the inhibitory effects of Erastin, a compound recognized for inducing cancer cell death and thereby attenuating tumor growth (Wu et al., 2020). Conversely, an increasing body of literature indicates that GSK-3β may display pro-tumorigenic properties in specific contexts. Investigations concerning HCC have revealed elevated expression levels of GSK-3β in tumor tissues, with higher levels correlating with poor patient prognosis (Zhang et al., 2020). In addition to its controversial role in oncology, GSK-3β is an integral enzyme that influences glucose metabolism. In the context of HCC, studies have shown that GSK-3β regulates glycolysis, supplying energy to rapidly proliferating cancer cells. The inhibition of GSK-3β activity has been associated with reduced glucose uptake, decreased lactate production, and diminished ATP levels in HCC cells. Moreover, such inhibition also results in the downregulation of key glycolytic enzymes, thereby affecting the overall metabolism in tumor cells (Fang et al., 2019). In our investigation, we observed a significant increase in the expression levels of GSK-3β in rats diagnosed with HCC. Notably, following the administration of genipin treatment, we recorded a reduction in the elevated GSK-3β levels. It is essential to emphasize that this study is groundbreaking in demonstrating the capacity of genipin to diminish GSK-3β expression, specifically within the context of HCC.

Fibronectin is a highly prevalent ECM protein that is essential for providing structural support and protection to cells within various tissues. In addition to its mechanical functions, fibronectin plays a critical role in multiple signal transduction processes that are fundamental to cellular communication and functionality. Notably, fibronectin is capable of undergoing alternative splicing of three specific exons from the same gene locus, which leads to the production of several isoforms (Yu et al., 2020). Among these isoforms, fibronectin extra domain A (FN-EDA) is particularly noteworthy due to its significant association with the development and progression of numerous tumors. Previous research has identified FN-EDA as a pivotal factor in promoting tumorigenesis through its influence on processes such as angiogenesis, lymphangiogenesis, and metastasis. This positions FN-EDA as a critical element in cancer progression and indicates its potential as a target for therapeutic intervention (Dong et al., 2024). Moreover, Toll-like receptor 4 (TLR4) has been recognized as an endogenous receptor that interacts explicitly with FN-EDA. TLR4 influences tumor growth through its effects on cancer metabolism and the immune response. Nonetheless, a gap in the current understanding exists regarding whether the FN-EDA/TLR4 signaling pathway contributes to the resistance of HCC cells to sorafenib (Liu et al., 2015). Investigating this relationship may provide valuable insights into the mechanisms underlying treatment resistance and unveil new approaches for enhancing the efficacy of cancer therapies. In our investigation, we observed a significant increase in the expression levels of fibronectin within the liver tissue of HCC rats. Notably, this elevated expression was substantially diminished following the administration of genipin to the HCC rats. It is important to illustrate that our study is the first to demonstrate the capacity of genipin to diminish fibronectin expression in HCC.

Our research findings have considerable implications for clinical applications. They suggest that they can be effectively integrated into various clinical investigations for several reasons. Previous studies have established that genipin is characterized by its low toxicity, rendering it a safe option for therapeutic use. In our recent study, we administered a precisely calibrated dose of 10 mg/kg to evaluate its effects. Notably, an additional investigation reported that when genipin was administered to laboratory rats at a higher dosage of 100 mg/kg, all rats achieved a remarkable survival rate of 100% (Hou et al., 2008). This finding emphasizes the compound’s excellent safety profile, even at elevated doses.

Conclusions

Our research findings indicate that genipin may represent a promising agent against cancer while simultaneously providing protective effects for hepatic function. This dual capability is primarily attributed to genipin’s ability to reduce free form of versican, frequently linked to tumor progression. Moreover, genipin appears to play a vital role in attenuating the intrinsic resistance of tumor cells. It accomplishes this by downregulating essential molecules involved in various signaling pathways that favor cancer cell survival and proliferation, EGF and its receptor EGFR, along with key downstream effectors such as PKB and GSK-3β. Through the modulation of the expression and activity of these molecules, genipin may disrupt the signals that facilitate tumor cell growth and survival. In addition to these mechanisms, genipin has demonstrated the capacity to reduce tumor cell survival rates by decreasing fibronectin levels.

Supplemental Information

Supplemental Information 1 Raw data

Supplemental Information 2 ARRIVE checklist

Supplemental Information 3 MIQE checklist

Additional Information and Declarations

Competing Interests

Author Contributions

Animal Ethics

Data Availability

The authors declare there are no competing interests.

Hanan M. Hassan performed the experiments, prepared figures and/or tables, and approved the final draft.

Mohammed Baradwan performed the experiments, prepared figures and/or tables, and approved the final draft.

Alaa Bagalagel analyzed the data, prepared figures and/or tables, and approved the final draft.

Reem Diri analyzed the data, authored or reviewed drafts of the article, and approved the final draft.

Ahmed Basilim analyzed the data, authored or reviewed drafts of the article, and approved the final draft.

Mohammed Z. Nasrullah performed the experiments, authored or reviewed drafts of the article, and approved the final draft.

Abdulhamid Althagafi analyzed the data, authored or reviewed drafts of the article, and approved the final draft.

Hussam I. Kutbi analyzed the data, authored or reviewed drafts of the article, and approved the final draft.

Abdulaziz A. Mohammed analyzed the data, authored or reviewed drafts of the article, and approved the final draft.

Hanan M. Alshan analyzed the data, authored or reviewed drafts of the article, and approved the final draft.

Mohammed M.H. Al-Gayyar conceived and designed the experiments, performed the experiments, prepared figures and/or tables, and approved the final draft.

The following information was supplied relating to ethical approvals (i.e., approving body and any reference numbers):

The Research Ethics Committee of the Faculty of Pharmacy at Delta University of Science and Technology approved the animal research protocol, assigning it the reference number FPDU8/2023.

The following information was supplied regarding data availability:

The raw data is available in the Supplemental Files.

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
