# Peer review of "Effect of modulating the extracellular matrix cross linkage by genipin on tumor cell resistance and survival in thioacetamide-induced hepatocellular carcinoma in rats"

_PeerJ, doi:10.7717/peerj.19680_

## Round 0.1 · original submission · Major Revisions

·

Basic reporting

The paper entitled "Effect of modulating the extracellular matrix cross linkage on tumor cell resistance and survival in thioacetamide-induced hepatocellular carcinoma in rats" shows the effects of genipin on the survival of rats with hepatocellular carcinoma and their survival. The findings are interesting. One suggestion is to reduce the adjectives and superlatives in the paper.

Experimental design

The experimental design is adequate but some pints must be addressed as the dilution of genipin since genipin is difficult to dissolve in water.

Validity of the findings

-

Additional comments

the results are adequate and the effects of genipin are impressive. Congratulations on this work.

Reviewer 2 ·

Basic reporting

This is an interesting study. Hassan et al reported that Genipin is a novel anticancer agen in thioacetamide-induced HCC model in rats.
Major concerns:
The authors claimed that Genipin exerts an anti-cancer effect by increasing the extracellular cross-linkage between hyaluronan and versican. However, in their study, they do not provide any experimental data to confirm it. Therefore, they shoud tone down their conclusions. genipin decreases HCC progression and improves HCC rats' survival by lowering the expression of EGF, EGFR, PKB, and GSK-3 and decreasing the expression of fibronectin.

Experimental design

OK.

Validity of the findings

Hassan et al reported that Genipin is a novel anticancer agen in thioacetamide-induced HCC model in rats.
Major concerns:
The authors claimed that Genipin exerts an anti-cancer effect by increasing the extracellular cross-linkage between hyaluronan and versican. However, in their study, they do not provide any experimental data to confirm it.

Additional comments

No.

---

## Round 0.2 · Minor Revisions

Firstly, I am sorry for the delay in handling this submission. We have lost contact with the original editor, so I have stepped in. I am satisfied with your response and the arguments you elaborated to the original set of comments. So, I ask that you address the following minor issues in what follows:

Please submit all the primary data associated with the study. Presently, there is no qPCR data which underpins the majority of the figures. This must be uploaded, and clearly annotated, and you should also make it clear how many technical replicates you have performed.

Once that is attended to, we can accept.

---

## Round 0.3 · accepted · Accept

Thank you for your prompt attention to these items. I am now recommending acceptance.